# Graphic Reconstruction of a Roman Mosaic with Scenes of the Abduction of Europa

Gregor Oštir [1], Dejana Javoršek [2], Primož Stergar [3], Tanja Nuša Kočevar [1], Aleksandra Nestorović [4] and Helena Gabrijelčič Tomc [1,*]

1  Department of Textiles, Graphic Arts and Design, Faculty of Natural Sciences and Engineering, University of Ljubljana, 1000 Ljubljana, Slovenia; go24773@student.uni-lj.si (G.O.); tanja.kocevar@ntf.uni-lj.si (T.N.K.)
2  DJN Studio, Consulting and Education, 1000 Ljubljana, Slovenia; dejana@dejana.si
3  Archaeological Research and Marketing of Cultural Heritage, Primož Stergar s.p., 3000 Celje, Slovenia; pstergar@gmail.com
4  Regional Museum Ptuj–Ormož, 2250 Ptuj, Slovenia; aleksandra.nestorovic@pmpo.si
*  Correspondence: helena.gabrijelcic@ntf.uni-lj.si; Tel.: +386-1200-3291

**Featured Application: The framework of the digital reconstruction of the Roman mosaic with scenes from the abduction of Europa and the inclusion of the digital reconstruction of the mosaic in an interpretive animation for the purpose of exhibition presentation.**

**Abstract:** This paper presents the reconstruction framework of the Roman mosaic with the central scene from the abduction of Europa. The mosaic depicting Europa, discovered in Ptuj (Slovenia) and dated from the second half of the third to the beginning of the fourth century AD, once decorated the representative room of a Roman villa. The experimental section addresses the materials and methods used in the 2D reconstruction of the mosaic, including the creation of line drawings of the mosaic based on the preserved part of the mosaic, photogrammetric acquisition, and the creation and processing of 1:1 raster reconstructions of the entire mosaic. This is followed by color management and interpretation approaches which allow the mosaic elements to be implemented in a 3D animation. The presented approaches could be implemented in the reconstruction process of other mosaics and archaeological objects with adaptations to the specifics of related objects.

**Keywords:** digital graphic reconstruction; cultural heritage; Roman mosaic; photogrammetry; the abduction of Europa

## 1. Introduction

In 1893, the architectural remains of roman villas with eight mosaics were discovered in Ptuj, Zgornji Breg. Ptuj today, once an ancient Roman colony of Poetovio, Slovenia, has developed into the biggest city between the northern Adriatic and Danube due to its strategic location. The city was founded in the 1st century AD (lat. Anno Domini) and reached its peak in the 2nd and 3rd century AD. At the beginning of the 5th century AD, it declined. Important state offices gave special importance to Poetovio [1]. The inhabitants of such an important city were able to afford luxurious architecture with luxury furnishings. One such wealthy district was located on the site of today's Zgornji Breg.

Mosaics of excellent quality adorned the villas, which connected the military and civilian parts of the city. The villa in which the mosaic of our study was located was never completely excavated. Consequently, we do not know exactly whether it belongs to the same architectural complex as the mosaics south of it. According to the equipment, this villa indicates the wealth and well-being of the owner. On the north side, it had a fireplace for central heating. From there, the heat was dissipated into the house spaces. The central representation space with mosaics had underfloor and wall heating, and was decorated with frescoes and stucco. On the floor was a multi-colored mosaic with the central motive

from the abduction of Europa [2], which may have adorned the reception room of a smaller villa. Due to the quality of the mosaic, it was stored in the Universalmuseum Joanneum in Graz, Austria and, today, it is exhibited in the museum's lapidarium. The mosaic with the motive from the Abduction Of Europa was cca. 400 × 400 cm in size [3]. Europa, sitting on a bull, is shown naked in a three-quarter view from the front. Behind her, drapery flutters. She holds the drapery with her right hand and the bull by the right horn with her left. The bull is shown in profile to the right in motion. Below are some floor lines illustrating water. The scene most likely represents the moment when the bull abducts the princess and runs off into the sea with her.

The mosaic is well enough preserved that it enables complete reconstruction. Figure 1 shows the mosaic of the abduction of Europa as exhibited in the museum in Graz (Austria). Although it is a floor mosaic, the remains of the mosaic are on the museum's wall, and a close-up view of the mosaic clearly shows the individual tesserae. It can be seen that, in the central part with the emblem representing Europa on a bull, the tesserae are much smaller than in the rest of the mosaic (Figure 2).

Stratigraphic data from the time when the mosaic was excavated are not so comprehensive to provide dating, so we must rely mainly on stylistic-typological analysis ([4], p. 456). The mosaic was most likely created between the second half of the third century and the beginning of the fourth century AD, and belongs to the so-called crisis period of the empire.

The myth of Europa has several variants and intertwines with other myths, and its basic structure is simple. The princess, who was gathering flowers with her companions on a meadow on the coast of Phoenicia, was abducted by Zeus in the form of a bull. As she climbed on his back he ran into the sea and carried her to Crete. From their union, five children were born. Thus, a new Cretan dynasty was formed. Europa and the children of Zeus inhabited the continent with their descendants ([5], pp. 167–169). In the context of a love story, idyllic piety, and prosperity, we can also understand the depiction of Europa on a bull from a mosaic found in the villa of a wealthy citizen of Poetovio. As a metaphor, the mosaic addressed visitors about the lifestyle, wealth, education, and values of the house master. These are connected with the values of the state, which, with its myths and traditions, is still based in paganism and represents a civilized world in contrast to the barbaric one ([4], p. 355).

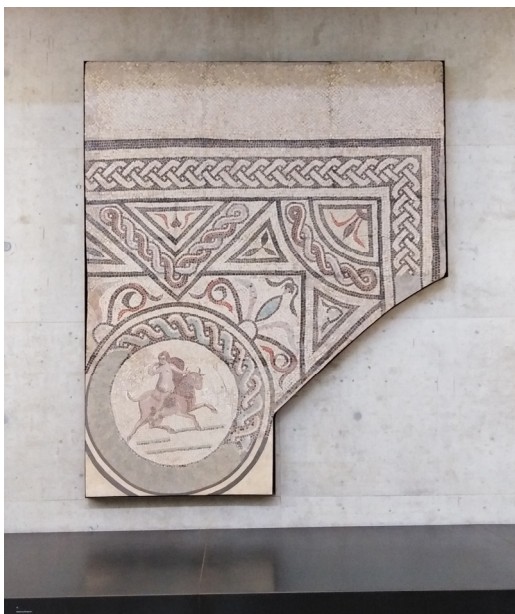

**Figure 1.** Mosaic with the scene of the abduction of Europa exhibited in the Universalmuseum Joanneum in Graz (photo: Barbara Porod).

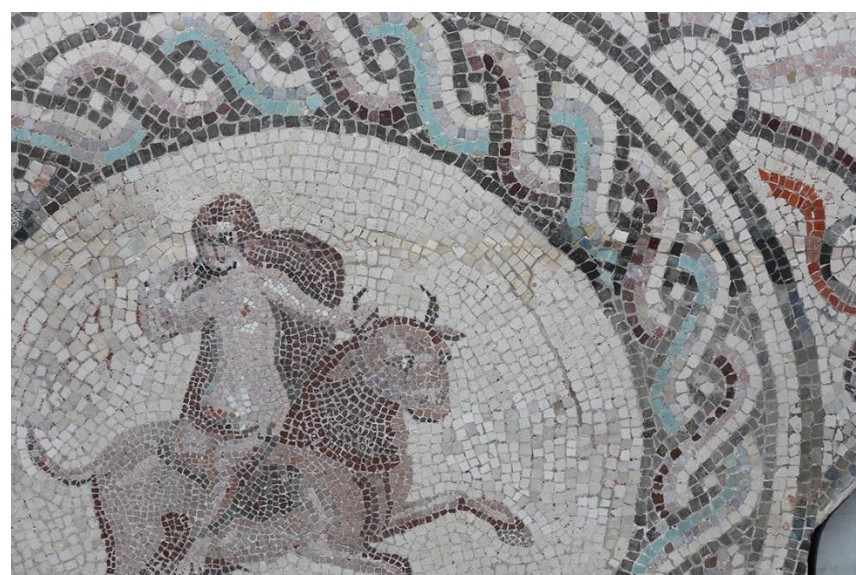

**Figure 2.** Close-up of the mosaic with the scene of the abduction of Europa (photo: Aleksandra Nestorović).

The mosaic with the scene of the abduction of Europa is one of the most interesting and suggestive ancient monuments found in Ptuj (Slovenia). The is not only because of the technical design of the mosaic and its quality, but also because of its message. The motif points to a long tradition of European culture and its wellbeing.

The goals of the presented research were as follows: metrically accurate reconstruction, color-accurate reproduction, and the interpretation of the mosaic in an interpretative 3D-animated presentation of the remains of an ancient Roman mosaic with the scene of the abduction of Europa, which was found in Ptuj, Zgornji Breg, and is exhibited in the Universalmuseum Joanneum in Graz (Austria). The Roman mosaic with the scene of the abduction of Europe is one of the most important mosaics found in the area of today's Slovenia, not only due to its quality, but also because of its still relevant message. The scene is connected to the myth of the colonization of the European continent. This is also why we decided to make a graphic reconstruction of the entire mosaic. In addition to digital reconstruction, the steps involved in the digital reconstruction of the mosaic were systematically described, taking into account all phases of mosaic processing that are important for understanding the reconstruction of the mosaic. The novelty of the research achieves historical and artistic insights specific to the colonization of the old continent, based on the technical work of reconstructing unpreserved sections of the mosaic (part of the main motif and the surrounded motives). Using archaeological evidence and records, the parts of the mosaic were digitally pieced together with the aim of understanding the original composition and artistic meaning of the symbolic abduction of a female (a Phoenician princess) by the supreme god (Zeus). This is a pioneering work for the Ptuj-Ormož Regional Museum and for the region as well. Additionally, the results of the research work comprehensively complement the other publications and studies with an example of the digital reconstruction of the mosaic with this scene.

Before starting the research, we posed the research question, "whether a complete reconstruction of the mosaic with the scene of the abduction of Europa, which is relatively well preserved and whose digital completion is predictable, can provide a visual experience, an interpretation in 3D and a starting point for further research and use with which the museum visitor can also interact".

The graphic reconstruction of the mosaic and the line drawing of the mosaic in a scale 1:1 serves as a museum database that we can use for the protection and presentation of cultural heritage. It can be used as a basis for the conservation and restoration processes of the mosaic, making replicas and for professional studies. The reconstruction can be used

for presentation purposes at an exhibition in museum spaces or online. The mosaic can be printed at a 1:1 ratio and it is suitable for further studies, virtual interactive presentations, and presentations in other media. It is also suitable for educational purposes. The user can imagine how the mosaic looked when it was in use and can place it virtually in an authentic space by enhancing it. Modern information and communication technology (ICT) enable the synergy of professional, technological, and interpretative communication skills and offers solutions in the virtual space. At the same time, we are graphically returning the mosaic to its original environment, where it was found, as it is stored abroad. The modern museum as a medium is particularly committed to conveying information and, above all, knowledge as a generally accessible social good. The direct and rapid accessibility of selected cultural heritage content is a great advantage for the user [6].

The results of the reconstruction were presented in an interinstitutional traveling exhibition, Women's Stories [7]. The exhibition was dedicated to the position of women in the past and a comparison with the present using museum objects. Thematically different vitrines with objects illustrated the status, gender, or social position of women in the past and present, from mythology and archaeology to modern times. Myths and stories were implemented to illustrate what women have gained or lost over the course of time. Through heritage and archetypes, we reflected modern women and their various social roles. In the context of the exhibition, we presented through the mosaic the myth of Europa. This work also aims to accurately determine the workflow for the graphic reconstruction of the mosaic, which could be used as a standard procedure for recording and reconstructing mosaics and beyond.

## 2. Examples of Mosaic Reconstruction

Digital reconstruction methods have been used for many years in the preservation and representation of cultural heritage, including mosaic reconstruction [8–10]. Traditionally, mosaic reconstructions relied on drawings and photographs for documentation. While laser scanning offered a promising 3D approach, including for mosaics, it resulted in low-quality mosaic images. To overcome this limitation, Structure from Motion (SfM) was introduced [11,12]. This technique utilizes regular cameras to generate high-quality 3D models with color information, proving to be a valuable tool for in-depth mosaic research. The results are high-resolution orthographic images, demonstrating SfM's effectiveness for detailed mosaic documentation.

The evolution of technology and the introduction of new software tools have significantly broadened the application of photogrammetry in the realm of cultural heritage, particularly in the documentation, analysis, and reconstruction of mosaics. These advancements have streamlined the photogrammetric process, incorporating algorithms from the field of computer vision to enhance automation and efficiency in creating 3D reconstructions from photographs. This shift towards more accessible and automated photogrammetric methods marks a notable advance in the preservation and study of mosaics, demonstrating the critical role of photogrammetry in the conservation of cultural heritage assets [11–13]. Utilizing SfM photogrammetry for mosaic preservation, we can create accurate 3D models of these artworks by capturing overlapping images from multiple viewpoints with just a standard compact camera, enhanced by georeferenced data for precise location mapping. Reference points are necessary for the orientation and placement of photographs in the spatial coordinate system. This step is critical for generating georeferenced and metric 3D models that accurately represent the mosaic's surface (cloud of points, orthophoto plan...). For effective computer processing and analysis, maintaining the spatial coherence of models is required from point cloud to orthophoto mosaics [14].

Photogrammetry is widely used to capture mosaics' data with a variety of specifications. The use of recording devices, such as drones, makes it possible to capture the image data of mosaics whose dimensions extend over large areas of antiquity [15], or to capture and process the details of mosaics through images, process them with software, and identify the materials used in the tesserae. In the study by Fioretti et al. [16], the

researchers created a digital record that allows for the identification of original sections and restored areas. Photogrammetry can be used in the analysis of tesserae used for recurring decorative patterns of the mosaic [17]. Photogrammetry not only aids the conservation of the mosaics, but also enables further studies directly on the digital model and provides valuable visual and metric data for interpretation and presentation. Visual information acquisitions of mosaics are useful in combination with 3D representations. A 3D model that combines mosaic details (shape, size, color) with informative data (materials, period) can improve the understanding of the site's history and the planning of future restoration and conservation [18]. Also, in restoration, the colors are important for data interpretation concerning the chromatic behavior of the materials [19].

Authors M. Monti and G. Maina [20] present a simple method for reconstructing mosaics that is not metrically accurate. The non-metric approach captures the mosaic without the use of photogrammetry or laser scanning. Digital acquisition was performed using an ordinary digital camera. For larger mosaics, a larger number of photos were taken with overlapping edges, which were then combined to obtain the high-resolution full size required to show the details of the mosaic and the print quality. During excavations in 2011, five rooms with mosaic floors were found in the square in Ravenna, probably dating from the early Roman imperial period (1st and 2nd century AD). The mosaics were removed for the purpose of restoration and musealization, but due to the large gaps they could not be integrated into the traditional restoration process without creating arbitrary reconstructions. Therefore, it was decided to digitally reconstruct the gaps in a non-invasive way without touching the mosaic. Simple and freely available software was used for the reconstruction. The goal of the digital workflow was to provide an example of virtual processing useful for conservators and restorers as well as scientists such as archaeologists and historians. They proved that this type of reconstruction can be performed without special knowledge of information technologies and computer skills. The reconstruction was carried out taking into account the geometric patterns that make up the preserved mosaic and which made the reassembly most reliable.

The results of image-based capture techniques, including close-range photogrammetry of mosaic parts and three-dimensional cultural heritage objects, are used in combination with augmented reality environments for interactive visualization [21]. Santachiara et al. [22] developed an augmented reality application (AR) that allows for the completion of missing mosaic areas by integrating the existing area and a virtual reconstruction based on a watercolor painting by Giuseppe Graziosi showing half of the mosaic with the missing parts. The existing parts of the mosaic were captured using the photogrammetric method, and the missing parts were digitally reconstructed using a high-resolution watercolor photograph. Half of the painted watercolor was mirrored to create a complete mosaic and then overlaid with an orthophoto of the original mosaic, cropping out the areas where the orthophoto of the mosaic overlapped with the watercolor photograph. The remaining portion of the watercolor photo was used in an AR application.

The relief of the mosaics is an important morphological aspect that is included in the reconstruction alongside the reconstruction of the mosaic colors. The inclusion of the relief is necessary if the representation of the reconstruction requires it, i.e., if the reconstruction needs a higher resolution and representation quality for further research purposes or a more accurate representation for the users. For the reconstruction of the relief, simpler approaches of using maps (bump and normal type) can be used, which in virtual representations only seemingly simulate the relief (at the level of optical phenomenon) or approaches that affect the geometry of digital models or the surface of mosaics (such as pie shift and DEM–digital elevation map) [23]. Tomography and ultrasound techniques are also used to reveal seemingly hidden subsurface details and deep patterns [24].

With the aim to reconstruct the Great Pitiunt Fortress, Glazov et al. [25] discovered that the mosaic floor in the fortress was the first to be reconstructed in order to ensure the correctness of the process. Only 10% of mosaic's remains were available that covered the interior area of the temple. The reconstruction of the mosaic was based on available

sources, excavation inventories, photogrammetry, aerial photographs of the remains of the temple, ortho maps, and DTM—digital terrain models. On the basis of initial references and photographs, smaller parts of the giant mosaic were reconstructed in plan in individual parts of the building. It was possible to reconstruct the motifs of the mosaic in parts where they were more uniform (uniform patterns) and their refinement was possible on the basis of studies on the construction of mosaics from the time of the dating of the mosaic. In their results, the authors state that the reconstruction of almost the entire mosaic was possible, as it did not contain very complex patterns. They also set out guidelines for the reconstruction of building mosaics from the period in question.

The reconstruction of the mosaic was also part of the research framework of the authors Fazio and Lo Brutto [26], who discovered that the mosaic covers part of the floor of the object under study, i.e., the complex of the Sanctuary of Isis in Lilybaeum, the ancient city of Marsala (southern Italy). The research uses a combination of digital and 3D approaches (2D reconstruction of the mosaic, 3D modelling) for the reconstruction of the mosaic and its placement in space alongside terrestrial laser scanners and photogrammetric surveys for the purpose of virtual reconstruction. A detailed and high-quality 2D reconstruction of the mosaic floor was possible using close-range photogrammetry, which provided the researchers with accurate size and color data. As the mosaic contained relatively simple, repetitive patterns, the reconstruction of the entire mosaic was not complex and was created by assembling basic motifs.

For the reconstruction of mosaics, automated methods are increasingly being used, which are possible with the help of image analysis for mosaics with simpler, repetitive motifs and patterns; for more complex mosaics, artificial intelligence (AI) methods can also be used. In recent years, AI-supported approaches have been developed with the aim of automating, accelerating, and qualitatively improving the analysis and reconstruction of mosaics. The prediction, classification, image completion, variation of image solutions, and automatic description of the different tiles (tesserae) are just some of the automation functions made possible by the deep learning-based method. These approaches are an important first step for the analysis of mosaics and enable better cataloguing and preservation of cultural heritage [27,28].

The authors Gil, Gomis, and Pérez [29] present a vectoral computer-aided approach for the automatic image analysis of mosaic images. The approach draws on group symmetry theory and perceptual psychology to extract information about the mosaic pattern, repetition, and arrangement. The authors divide the approach into three main phases: restore, the purpose of which is to obtain data about the missing parts of the mosaic, unify, a phase in which a decision is made between different motifs; and standardize, in which the motifs are positioned exactly on the axis of symmetry and the center of rotation. The method is suitable for mosaics with simple, repetitive motifs.

Brutto and Dardanelli [30] investigated the efficacy of photogrammetry and computer vision for high-uncertainty 3D reconstruction in archaeology, particularly for mosaics requiring high-precision measurements (sub-millimeter). The authors evaluate this approach by capturing 3D data from three mosaics of varying size and location in Italian museums. The goal was to assess the potential and limitations of the technique, specifically regarding camera calibration, for creating detailed 3D models and full-scale orthographic images for documentation and restoration purposes.

Research by Moral-Andrés et al. [31] proves that AI approaches are increasingly relevant for inclusion in the framework of mosaic reconstruction. Research recently analysed the Dall-E tool with the aim of reconstructing images of both mosaics with more complex motifs (people, animals, etc.) and mosaic pieces with simpler repetitive elements (patterns). The tool proved to be reliable in providing reconstruction solutions as it adequately completed parts of the mosaic images where parts of human bodies, animal bodies, and depictions of the interactions between the motifs of humans and animals were missing. The tool was particularly successful in reconstructing and completing simple repetitive patterns. The conclusion of the study is also that the use of the tool during the study period (2022)

does not reach the level of manual editing and reconstruction of mosaics. The shortcomings lie mainly in incorrect or inaccurate interpretations and representations of the position of human body parts, the additions of incorrect elements (a clothed body instead of a naked one), and the addition of elements in nonsensical places (for example, additional limbs of animals, etc.). In any case, the authors conclude that the current AI-supported process can help with possible interpretations of the missing parts of the mosaic, on the basis of which the researchers then decide on the final solution.

## 3. Materials and Methods

The experimental part of the graphic reconstruction of the mosaic was carried out by a multidisciplinary team. The team consisted of experts in archaeology, photogrammetric acquisition, graphic design, visualization, and color management, who determined the workflow of the reconstruction. The reconstruction workflow was divided in five phases: 1. preliminary studies; 2. photogrammetry; 3. raster reconstruction; 4. color management and printing and 5. 3D-interpretative animation, as is with the sub-phases presented in Figure 3.

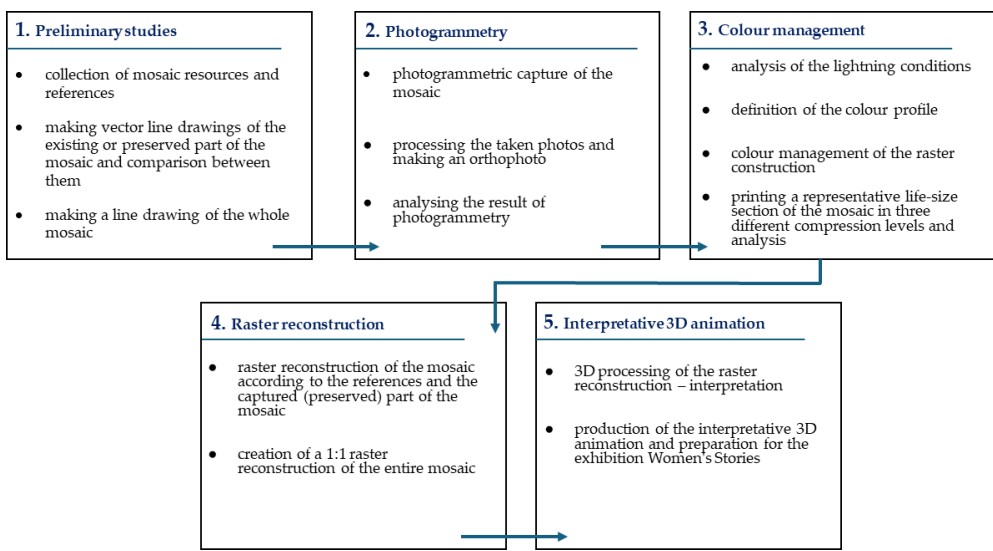

**Figure 3.** Reconstruction workflow.

### 3.1. Preliminary Studies

The analysis of the surviving pictorial and written sources on the mosaic was carried out by an expert in the archaeology of the Roman period. The sources were evaluated qualitatively with regard to their archaeological relevance, using information about the time of the mosaic's creation (social, residential, spatial organization) and the starting points of the mosaic's creation. Sources were preserved as part of the original mosaic, the database InterArch-Steiermark [32], the database of the Universalmuseum Joanneum (hand drawing of the mosaic on foil in the ratio 1:1), publications of the mosaic with photos, and drawings and floor plan of the architecture ([1,2], pp. 4–5; [3] pp. 599, 601–602).

The vector drawing of the mosaic was based on a reference mosaic line drawing reconstruction (Figure 4, upper left) and on a mosaic scheme of the ideal baselines of the entire mosaic according to Djurić (Figure 4, upper right) [3]. The photograph was first aligned to scale with reference points placed at a measured distance of one meter. A comparative analysis was made between the ideal edges of the mosaic and the contours from the foil. The existing linear outlines of the mosaic, which are stored on a transparent foil (Figure 4, bottom left), served as the first tool for reconstruction with the mosaic's measurements 1:1 as vector data using the marked angles and dimensions drawn on the sketch (Figure 4, bottom right). Line vector reconstruction, without details (Figure 4, bottom right), and vector drawing of the entire mosaic, with the details (Figure 5), were drawn in

Adobe Illustrator (version 25.2). The linear reconstruction was carried out in the first phase in order to improve the museum documentation (graphically with bordures and filling motifs), as the existing documentation before this research only contained a linear scheme without details (by Nejka Uršič, Figure 4) and a schematic representation (by Bojan Djurić, Figure 4). In addition, the mosaic was inaccessible at the time this research began.

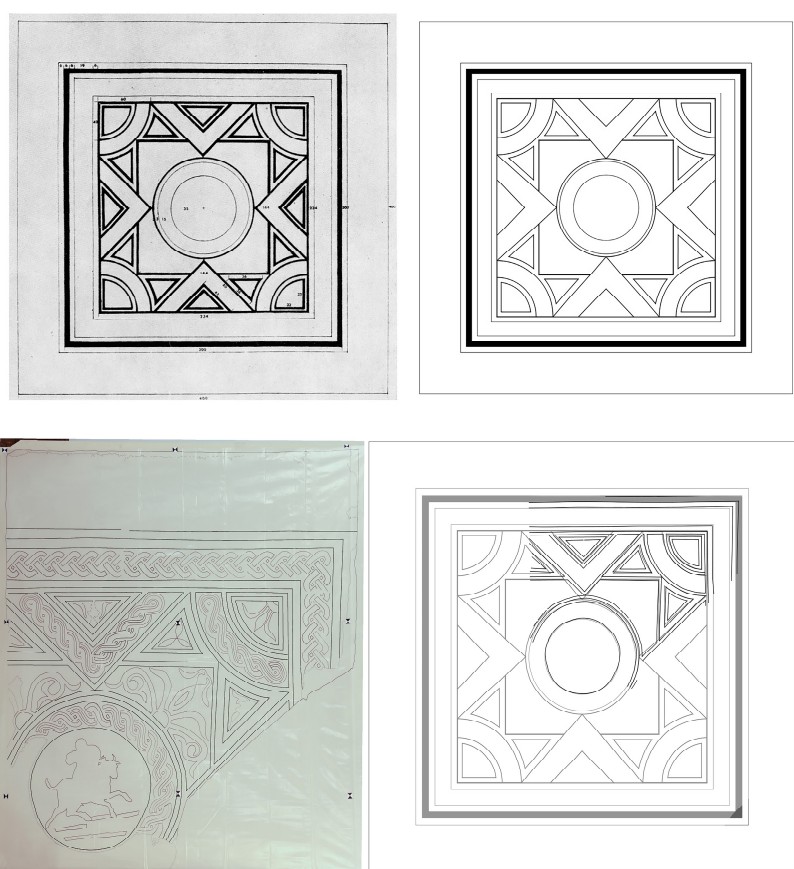

**Figure 4.** Reference mosaic line drawing reconstruction (**upper left**) by Nejka Uršič and reconstructed vector plot (**upper right**) after Djurić, outlines of the preserved mosaic on the foil in actual dimensions (**bottom left**) and the process of vector drawing based on the notes on the foil 1:1 by Nejka Uršič (**bottom right**).

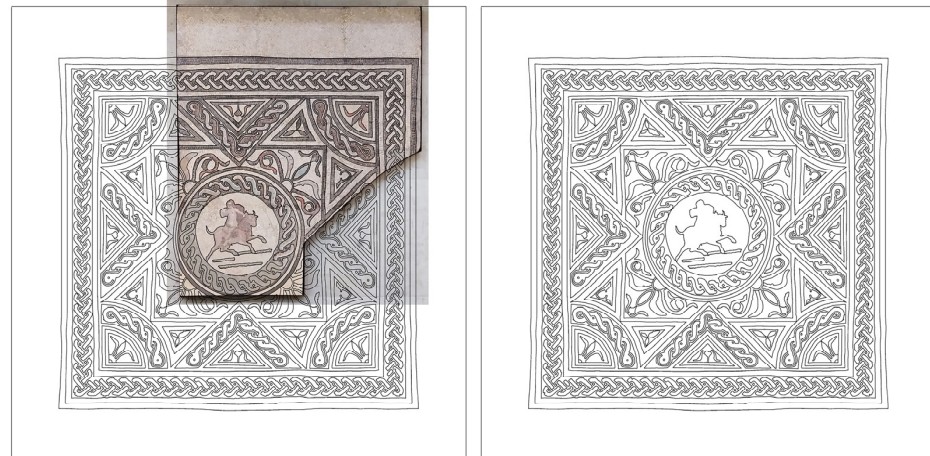

**Figure 5.** Vector drawing of the entire mosaic with a superimposed photo of the original (**left**) and details (**right**).

### 3.2. Photogrammetry

The mosaic was photogrammetrically recorded and an orthophoto was created. The recording technique was close range photogrammetry, according to the SfM method.

A Canon EOS 70D DSLR camera was used for the acquisition of the photos in the RAW file format, and a Leica electronic tachymeter was used to accurately measure the reference points (GCPs) placed directly outside the mosaic (Figure 6). The camera settings were as follows F: 3.51, 1/16 s, ISO value 100.

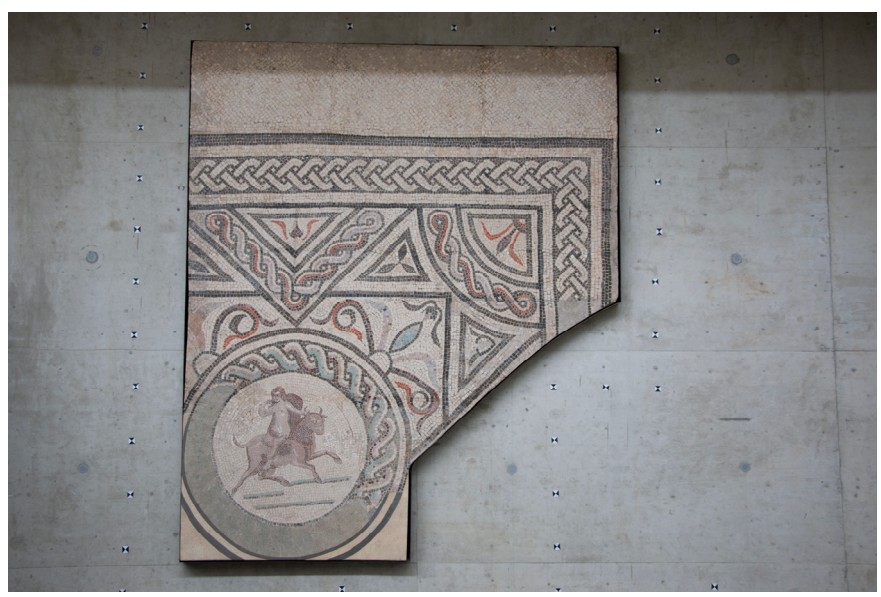

**Figure 6.** Reference points around the mosaic.

The photographs were taken in a square grid with average distance of 22 cm in between the individual photographs. The distance from the surface of the mosaic was approximately 1.2 m. The goal was to achieve 80% photograph overlap and to capture as much detail as possible. In order to reduce scale error, 21 Ground Control Points (GCP) with defined values x, y and z (Figure 6), were placed around the mosaic on the surface of the wall on which the mosaic is placed. The numbering of the points started in the bottom-left corner outside the mosaic and continued to the right in an anti-clockwise direction [33–35]. Each GCP was measured using Geodetic Total Station Leica TS09plus 5″ (Leica Geosystems AG, Heerbrugg, Switzerland). The measurements were made in reflector less mode using a visible red laser beam. The distance of placement of the total station from the mosaic and GCP was approximately 6 m. The measurements were made in the local coordinates system. The accuracy of the measurements was better than 0.5 cm. This allowed us to orient the mosaic correctly and to set up a coordinate system that allows reconstruction in its actual dimensions.

Agisoft Metashape 2.0.0 software was used for the processing or photogrammetric process. Of the 253 photographs captured, 206 photographs were used for this step, without the test plate. The first step was to create a sparse point cloud and to perform the referencing of the captured photographs to the measurements of the set reference points. The exact positions of the points had to be marked in the software on each photo used for processing. This was the most time-consuming or labor-intensive process in the processing phase, but the accuracy of the model depended on it. This was followed by cleaning and optimizing the sparse point cloud. After model optimization with 14 out of 21 GCP used, the average 0.0014 m (1.4 mm) error of the mode was achieved and the achieved Ground Sampling Distance (GSD) in pixel size was 0.000388981 m. The next step was to create a dense point cloud. A DEM (Digital Elevation Model) was then produced and the final step in this phase was the production of an orthophoto mosaic (Figure 7). The process was first carried

out with the photographs without the color profile attached, and then repeated with the photographs with the color profile attached.

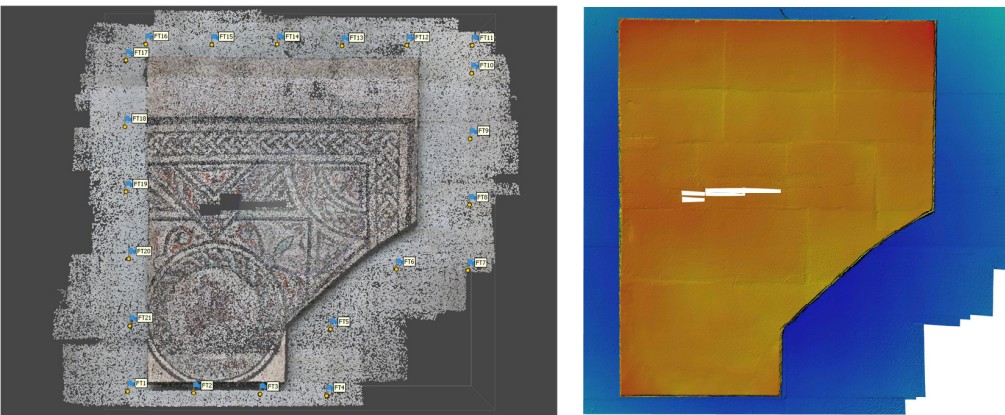

**Figure 7.** Generated dense point cloud from photos in Agisoft Metashape (**left**) and created digital elevation map (DEM) in Agisoft Metashape (**right**) which displays the different rectangular surfaces of the mosaic, probably after the restoration intervention of the relocation of the mosaic.

*3.3. Color Management*

An analysis of the basic hues of the colors appearing in the mosaic was carried out to identify five representative colors. We found that in the central part of the mosaic there are more hues of color, which give a visual sense of plasticity to the figures, while the surrounding part is simpler. In addition, there is also a difference in the size of the mosaic tesserae. In the central part with more detail, the tesserae are much smaller than otherwise. The color values were covered according to the possibilities of intervention in the mosaic i1Pro 2, X-Rite) and according to the references about the colors of the mosaic [3]. The representative colors of the mosaic are color 1 white ($CIE_{Lab}$ = 61.245, −0.945, 5.014); color 2 grey ($CIE_{Lab}$ = 52.612, 1.432, −3.792); color 3 black and dark grey ($CIE_{Lab}$ = 11.561, 1.772, −3.226 and $CIE_{Lab}$ = 22.482, 2.210, −5.609); color 4 red ($CIE_{Lab}$ = 26.531, 33.091, 22.630); and color 5 blue-green ($CIE_{Lab}$ = 50.279, −10.994, −12.264), as presented in Figure 8.

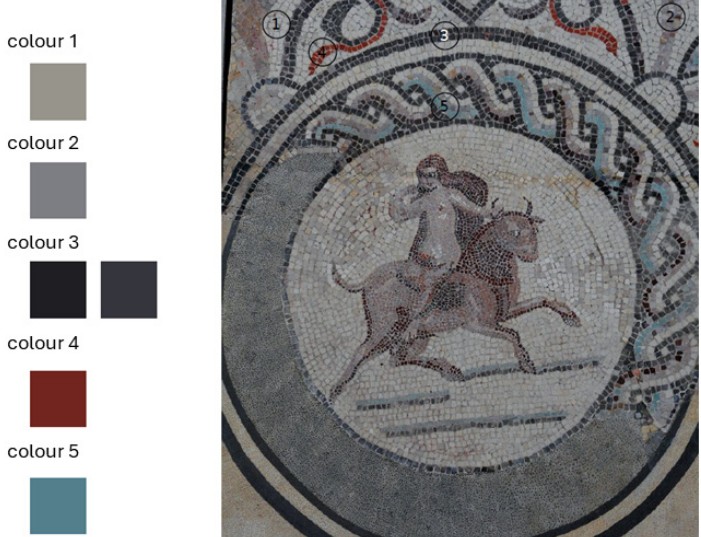

**Figure 8.** Representative colors of the mosaic.

The model was created using 253 photos and a color test chart ColourChecker DC (X-Rite) with 237 color patches, which were photographed with a digital camera and the settings as mentioned in Section 3.2. Photogrammetry. The test chart was placed in front

of the mosaic under the same lightening conditions after photographing each section of the mosaic (Figure 9). It was also important to take the photographs in RAW file format as this format allowed for the extraction and the creation of a color profile. For uniform illumination, two lights (Lupo Superpanel Dual-Color LED studio light) were placed at 45 degrees, relative to the perpendicular surface of the mosaic, with diffused light to avoid shadows. In regard to the settings on the camera, the ISO was kept as low as possible to slightly overexpose the image.

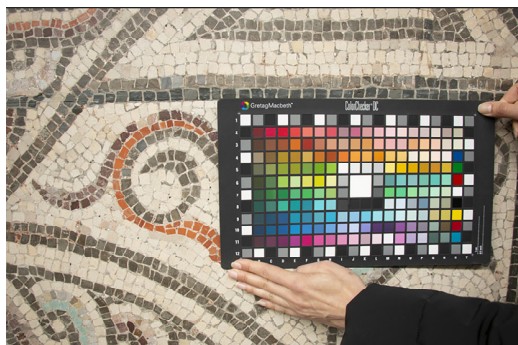 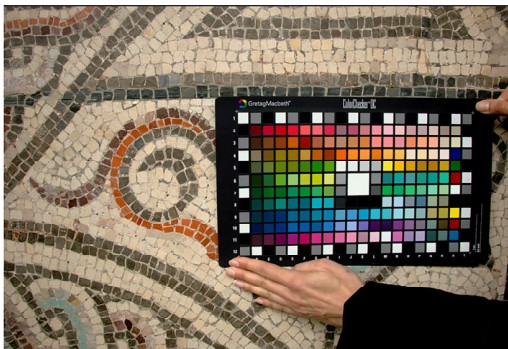

**Figure 9.** Comparison of photos of the mosaic without (**left**) and with the attached color profile (**right**) (photo: Primož Stergar).

RAW images were converted to TIFF files using Adobe Photoshop (version 22.2), and the settings defined the color temperature at 5900 K. The color profile of the digital camera was created according to the measurements on the color test chart in an open source program Argyll with tool named scanin, which converts a TIFF image of a test chart into .ti3 device values (RGB) and a tool named colprof which creates a color profile from .ti3 file. The created color profile of the digital camera was assigned to 253 photos in Adobe Photoshop 22.2.

### 3.4. Raster Reconstruction of the Complete Mosaic

The resulting orthophoto prepared with photogrammetry was the basis for a raster reconstruction of the entire mosaic. As the original orthophoto was very large, we decided to reduce the file size to a size that was suitable for the work and, at the same time, had enough visible details. We decided to use the orthophoto in JPG quality 8 format (compression performed in Adobe Photoshop 22.2), as there were no visible differences between the original and the compressed version when viewing the mosaic from the perspective of a visitor. The final file used to reconstruct the mosaic in its entirety was 60 MB in size.

The raster reconstruction was carried out by mapping the data from the preserved parts of the mosaic to the parts where the mosaic was missing. Just enough of the mosaic was preserved to make out all the geometric data and to predict the appearance of the rest of the mosaic. When mapping the preserved part, it was questioned whether the parts that would be copied might look too symmetrical. In this case, further corrections would be necessary. The reconstruction was carried out using reversible editing masks and editing tools (Figure 10). The mosaic was first copied and mirrored across the horizontal axis, trying to achieve a match. Using the mask, we deleted the parts that overlapped the original mosaic. We continued with similar mapping and transformation operations.

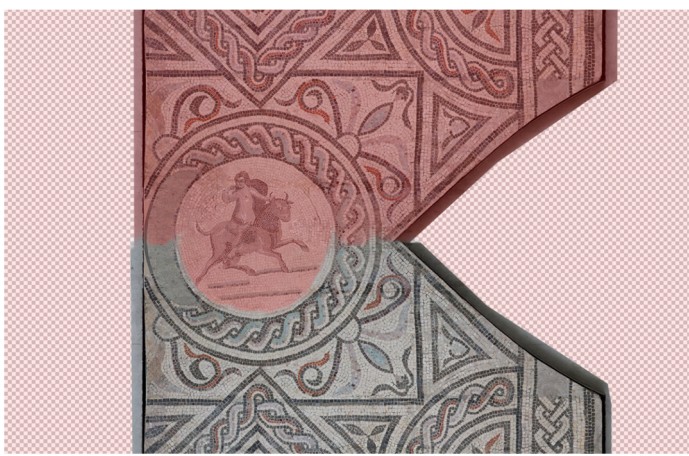

**Figure 10.** Example of mapping and attempt to match the original.

After mapping the parts and making adjustments between them, we made further corrections to the parts of the mosaic, trying to make it as close to the original as possible. The outermost edge of the mosaic, which was mirrored from the upper part of the original mosaic and was darker than the rest of the mosaic due to its position in the shadow, was "faded" by cloning data from similar parts. As we were not satisfied with the unevenness of the extreme black edge of the mosaic on the mapped parts, we straightened the edge on these parts.

### 3.5. Interpretative 3D Animation

The results of the reconstruction and its phases were summarized in an interpretive video and 3D animation, which was created with Blender 3.3 (BlenderFoundation, Amsterdam, The Netherlands) and Premiere Pro 23.0 (Adobe Systems, San Jose, CA, USA). The image results of the creation of the line drawing (contour of the mosaic), line drawings of details within the mosaic, individual raster, and color layers, by which the process of joining the raster reconstruction can be seen, were included in the interpretative animation. The final part of the animation was the artistic approach to the interpretation of the tesserae mosaic using the method of animating a system of particles whose movement along the path were determined using representative shapes of the mosaic motives.

### 4. Results and Discussion

#### 4.1. Raster Reconstruction

Key steps of raster reconstruction are presented in Figure 11. The result is a digital graphic reconstruction of the complete mosaic of the abduction of Europa, including the missing parts, as shown in Figure 12. Most of the mosaic is based on geometric patterns that are typical of the period in which the mosaic was made and are repeated evenly throughout the mosaic. The mosaic was relatively well preserved, with preserved essential elements of the mosaic, such as borders, figures, repeating pattern fields, which facilitated the transfer of pictorial information and preserved surfaces to non-preserved ones, with the aim of completing it as authentically as possible.

Since the mosaic scheme is based on the perfect symmetry of the motifs (Figure 11a), initially, the right side of the mosaic was reconstructed by mapping the upper (original) half (Figure 11b). One of the four triangles, whose vertices touch the inner circle, was supplemented and for the frame border with a multi-colored double strand guilloche, the black band and a white frame band were added (Figure 11c,d). When this part was completed, more than half of the mosaic was reconstructed (Figure 11e). Then, the preserved part of the mosaic on the left side was mapped (Figure 11f), and the missing triangle was filled the same way as the one on the right side (Figure 11g–i). At the same time, the missing part of the multi-colored double strand guilloche of the central field with a figural scene

was added (Figure 11g). The final product was improved, and the errors in the mosaic were corrected that would otherwise have been repeated. We also cleaned up the damage on the white frame band (Figure 11j–l).

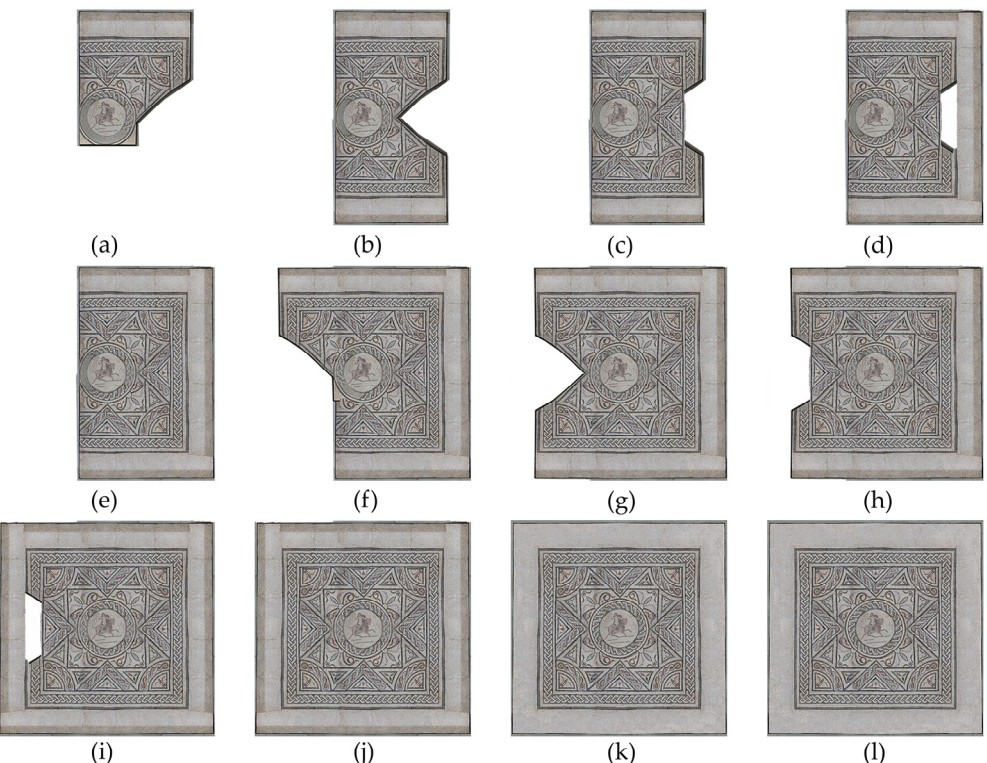

**Figure 11.** Key steps of raster reconstruction, (**a**) upper original half of the mosaic, (**b**) reconstruction of the right side, (**c**) added triangle (one of the four), (**d**) added frame border with a multi-colored doublestrand guilloche, the black band and a white frame band, (**e**) reconstructed right part of the mosaic, (**f**) mapping of the preserved part of the mosaic on the left side, (**g**–**i**) reworking of the left part with the addition of the missing triangle and the missing part of the multi-colored doublestrand guilloche of the central field with a figurative scene, (**j**) improvement of the final product, (**k**,**l**) cleaning up the damage on the white frame band.

This work was progressive. Each step of the digital addition made it possible to think about the next step, both from the archaeological point of view to understand the meaning of each component of the mosaic, and from the point of view of graphic interventions and editing of the image information with the aim of achieving the visual communication of the mosaic. During the work process, we had the greatest challenges in identifying the darker band in the upper part after we had assembled the reconstruction of the photogrammetric images. We realized that, due to the lighting (light through the windows) in the museum where the mosaic is exhibited, this upper part is regularly exposed to slight light influences and is, therefore, also slightly darker. In addition, in the final reconstruction, when visualizing the entire mosaic, digital image processing interventions are known to occur in areas with patches of visual artifacts that are unevenly distributed across the surface of the mosaic and are more noticeable in areas where there are fewer colored elements of the mosaic. Figure 13 shows on the left, a shadow "cast" on the mosaic by the ceiling of the museum, on the surface of which we discovered a darker band of tesserae even after additional illumination of the mosaic, and on the right, visual artifacts on the surface of the lower left part of the mosaic, which was reconstructed entirely from the image information of the preserved parts.

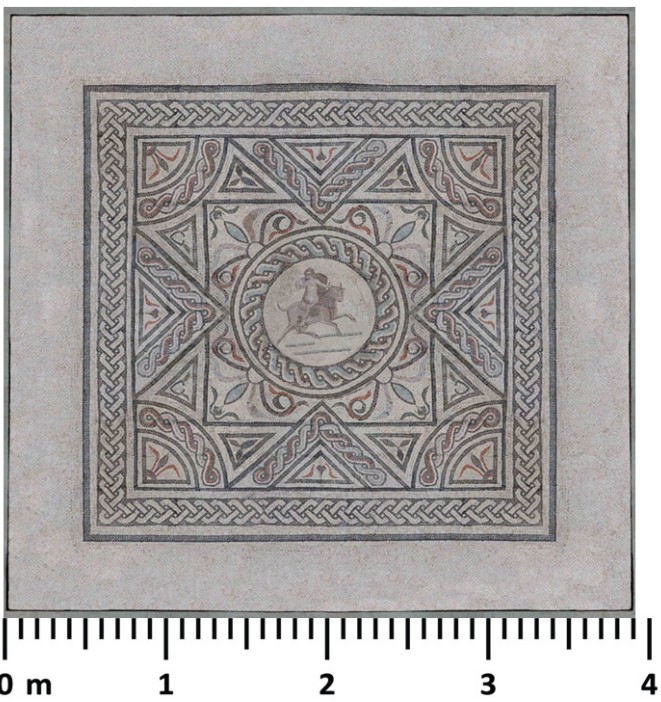

**Figure 12.** The final digital graphic reconstruction of the entire mosaic.

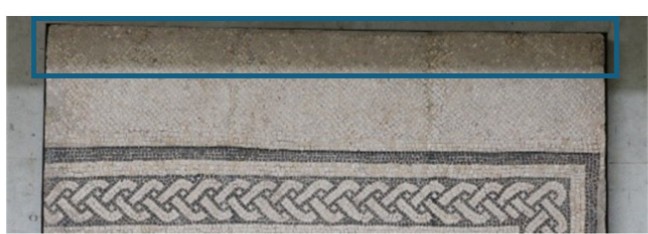

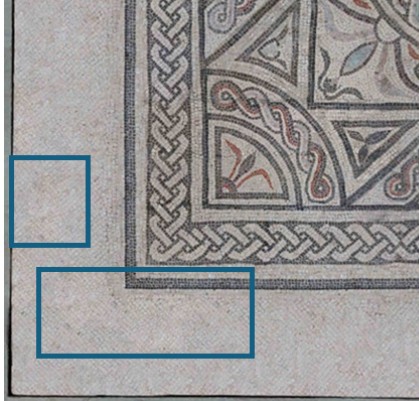

**Figure 13.** A darker band of tesserae probably caused by the continuous shadow casting on the mosaic by the ceiling of the museum (**left,** marked with blue box) and visual artifacts on the surface of the lower left part of the mosaic (**right,** marked with blue box).

The reconstruction of the mosaic serves the preservation of the monument, further studies, and the visitor experience. The reconstruction with the data acquisition on a scale of 1:1 provides information on how the tesserae are laid and their position in the whole mosaic contributes to the understanding of the mosaic production process and complements the museum's documentation. During installation, the mosaics were cut into pieces several times and then reassembled. In the process, the mosaic stones could be damaged or lost. In pieces, the mosaics could also be mixed up or damaged. The reconstruction, therefore, serves as a database for mosaic reconstruction and for the possible creation of a replica. Also, the colors are important for t data interpretation concerning the chromatic behavior of the materials in restoration process. In general, such databases are provided for the preservation of cultural heritage. This is why metadata is also very important in archaeology and heritage conservation. The graphic reconstruction of mosaics can also be used in the context of ancient villa architecture. The placement of the mosaic, together with the reconstruction of frescoes and stucco found in the same room, suggests the hypothetical

use of the space according to the composition and ornamentation of the mosaic. Due to the orientation of the central motif, the entrance to the room was most likely from the north side. Finally, the mosaic reconstruction is important for the visitor's visual experience. The graphical reconstruction can be used in the museum's exhibitions and other museum media. The visitor can also handle the virtual item.

### 4.2. Interpretative 3D Animation

The interpretative animation aimed the presentation of the technical approaches towards reconstruction, work phases, and final digital reconstruction for the participants of the international traveling exhibition in an interactive and attractive way (Figure 14a–f). The individual frames of the animation show the step-by-step reconstruction of the mosaic, and thus, also the essential design and structural elements of the mosaic, its layers, motifs, and structural components. In our opinion, such a systematic presentation of the working layers of the reconstruction would not be possible if fully automated and algorithmically controlled processes were used in the reconstruction. In this way, the digital composition of the mosaic was brought closer to the exhibition visitors in a visual and tangible way.

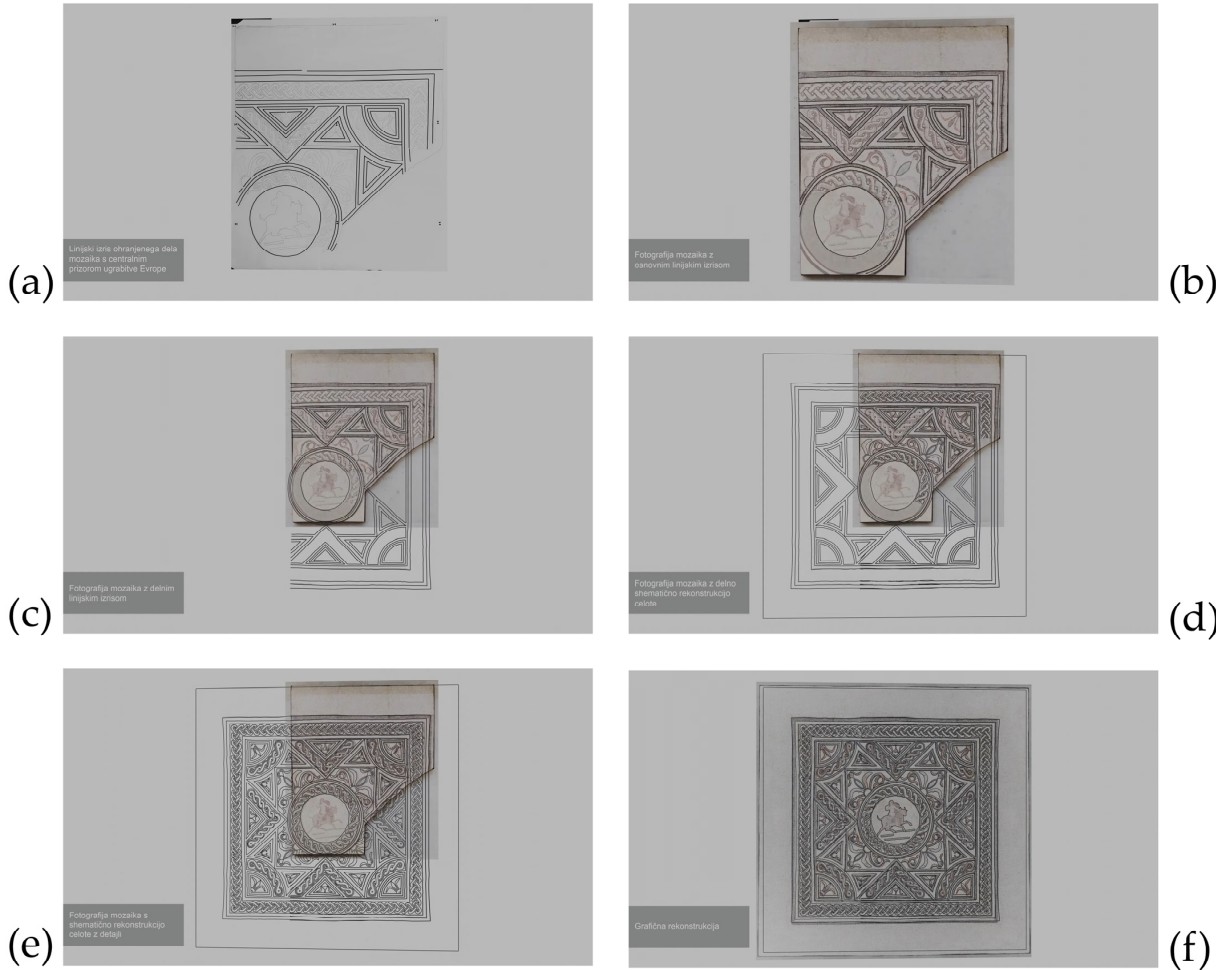

**Figure 14.** Video frames of the interpretative animation with the reconstruction procedure: (**a**) line drawing of the preserved part of the mosaic with the central scene of the abduction of Europa; (**b**) a photo of the mosaic with a simple line drawing; (**c**) a photo of the mosaic with a partially completed line drawing; (**d**) a photo of the mosaic with completed simple line drawing; (**e**) a photo of the mosaic with a schematic reconstruction of the details; (**f**) schematic reconstruction of the mosaic with the central scene of the abduction of Europa, the whole with all details (author of illustrations and color graphic reproduction: Gregor Oštir, author of the video: Anže Mrak).

Figure 15 shows frames with the interpretative 3D animation (camera movement, path movement) and the procedural 3D approach (particles). In this part of the animation, the artistic interpretation of the authors was included by integrating basic and procedural animation principles. The visualization shows the revival of tesserae, which emerge as a group of objects–tesserae (particles)–from the base of the mosaic and represent, in volume, the basic motif of the mosaic, namely the scene of the abduction of Europa.

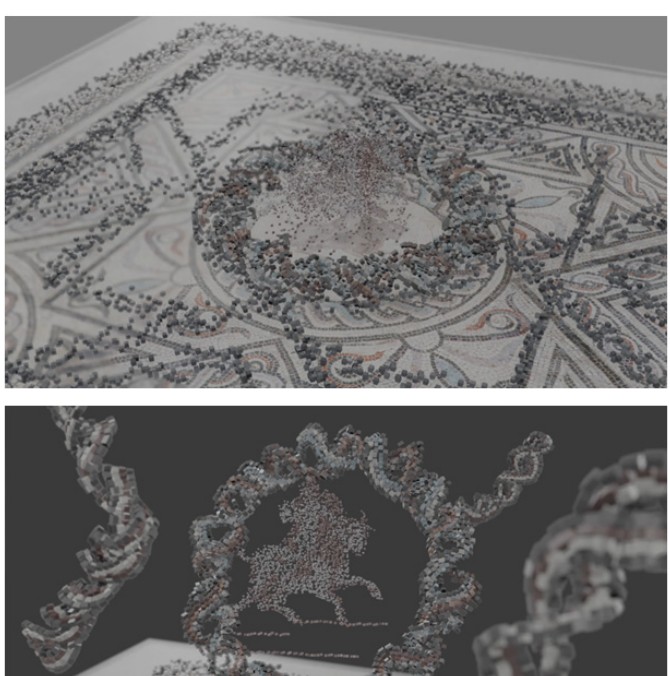

**Figure 15.** Video frames with the interpretative 3D animation—(**top**), and 3D procedural approach with particles—(**bottom**) (author of illustrations and color graphic reproduction: Gregor Oštir, video author: Anže Mrak).

## 5. Conclusions

The graphic reconstruction of Roman mosaics represents an important contribution to the preservation of cultural heritage. The result of this work is a digital graphic reconstruction in full size, which can serve a variety of purposes and allows unlimited possibilities for digital manipulation without invasive interventions in the original.

This research presents a novel approach to gaining historical and artistic insights into the colonization of Europe, focusing on the symbolic motif of abduction. Through the technical feat of digitally reconstructing missing parts of a mosaic (including the central motif and surrounding elements), the study utilizes archaeological evidence and records. This digital reconstruction aims to shed light on the original composition and artistic meaning behind the depiction of a symbolic abduction, which probably represents the abduction of a Phoenician princess by Zeus. This work represents a pioneering achievement for the Ptuj-Ormož Regional Museum and the region, as it provides a unique example of digital mosaic reconstruction for further study. In addition, the research results comprehensively complement existing publications and studies by providing a concrete example of a digitally reconstructed scene with this particular mythological theme.

Based on the experience of working with museum materials and databases, the graphic reconstruction of the mosaic serves the preservation of the monument and, beyond that, the preservation of cultural heritage, further studies, and the visitor experience. It contributes to the understanding of the mosaic making process and complements the museum

documentation, serving as a database for the mosaic reconstruction and for the possible creation of a replica. It can also contribute to a better understanding of Roman architecture and can be further used in the original virtual environment. Finally, it enhances the visual experience and visitors' understanding of the ancient mosaic and architecture. The reconstruction can be used for presentation purposes at an exhibition in museum spaces or online. The mosaic can be printed at a 1:1 ratio, and it is suitable for further studies, virtual interactive presentations, and presentations in other media. The user can imagine what the mosaic looked like when it was in use and can place it virtually in an authentic space by enhancing it. At the same time, we are returning the mosaic to its original environment, where it was found, as it is stored abroad. As such, we present it to the local population and the museum to completes its database and collection. This way, we also enable the direct and rapid accessibility of cultural heritage content of mosaic.

Based on the aforementioned, we answer affirmatively to the research question stated at the beginning of the research. The research of existing sources (written and pictorial) about the mosaic and the digital reconstruction of the mosaic in 1:1 scale enables a comprehensive visual experience, an interpretation in interactive 3D animation with the aim of using it for exhibition purposes, and acts as a starting point for further archaeological research.

This project demonstrated the benefits of multidisciplinary collaboration. Experts from different disciplines were involved, each contributing knowledge and experience from their field. The work took place in several planned phases, which were adjusted according to needs and newly acquired information. The presented framework for research and digital reconstruction, with extensions based on the specifics of other mosaics, could become a standard procedure that would simplify and unify the implementation of graphic reconstruction of mosaics and other archaeological objects. Of course, each mosaic is a case in itself, mainly because of the different degree of preservation, but if we can identify with certainty the geometric patterns on the preserved part of the mosaic and their continuation in the missing parts, a similar procedure can be used for reconstruction. Since it is a non-invasive and non-destructive reconstruction approach, the method could be used more frequently in the field of cultural heritage preservation in the future.

The presented study opens up possibilities for further research and deepening of the understanding of the meaning of the digital reconstruction of the mosaic. We are currently exploring the possibility of supplementing the methodology with AI, which we have used in the reconstruction of other mosaics, but which did not provide satisfactory results. The AI tools required an extremely large number of explanatory instructions, but these did not help to adequately fill in the missing parts of the mosaic. Artificial intelligence tools could certainly enrich the reconstruction workflow, but only if sufficient mosaic data is available. In our experience, for more complex motifs and unpredictable sequences, AI tools do not achieve the correctness and complexity of supplementing the image templates to the starting points of the remains of mosaics [36].

The colors are important for the data interpretation concerning the chromatic behavior of the materials in restoration process. Moreover, improving the color management process could be performed even more specifically at the level of determining the colors of individual groups of tesserae belonging to a particular tesserae type. This would be based on the color palettes used by geologists by taking into account the stone types. For this procedure, the geologist would have to determine the types of tesserae. Determining the origin would indicate the origin of the stones, and thus, also represent the process of mosaic production. We estimated that this process would be very time-consuming and costly to implement, which, in our opinion, would not outweigh the quality and accuracy of the final digital reproduction for the purpose of presentation at the exhibition. However, it would possibly aim to improve the quality of further archaeological studies of the mosaic.

**Author Contributions:** Conceptualization, A.N., H.G.T. and T.N.K.; methodology, A.N., H.G.T. and T.N.K.; software, G.O., H.G.T. and T.N.K.; validation: D.J. and A.N.; formal analysis, H.G.T. and T.N.K.; investigation, P.S., H.G.T. and T.N.K.; data curation: G.O. and A.N.; writing—original draft preparation, G.O. and H.G.T.; writing—review and editing, A.N., H.G.T. and T.N.K.; visualization, G.O., H.G.T. and T.N.K.; supervision, H.G.T. and A.N.; funding acquisition, H.G.T. All authors have read and agreed to the published version of the manuscript.

**Funding:** The authors acknowledge the financial support from the Slovenian Research Agency (research core funding No. P2-0450).

**Institutional Review Board Statement:** Not applicable.

**Informed Consent Statement:** Not applicable.

**Data Availability Statement:** Data are contained within the article.

**Acknowledgments:** Sincere thanks to Barbara Porod for the photos of the mosaic and to Anže Mrak for the production of the 3D animation of the mosaic.

**Conflicts of Interest:** Author Dejana Javoršek was employed by the DJN Studio, Consulting and Education. Author Primož Stergar was employed by the Archaeological Research and Marketing of Cultural Heritage. Author Aleksandra Nestorović was employed by the Regional Museum Ptuj–Ormož. The remaining authors declare that the research was conducted in the absence of any commercial or financial relationships that could be construed as a potential conflict of interest.

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
