# Peer review of "Graphic Reconstruction of a Roman Mosaic with Scenes of the Abduction of Europa"

_applsci, doi:10.3390/app14093931_

Round 1

Reviewer 1 Report

Comments and Suggestions for Authors

The article is relevant and interesting. Revision has to be done for the paper to publish it. I would really appreciate if the authors adjust their paper in the specified places.

The beginning presents the historical context but could enhance the introductory part with a clearer connection of the significance of the mosaic's digital reconstruction and its addition to current research and technology in the field. In addition, some other reconstruction method are suggested (https:(https://doi.org/10.1016/j.enggeo.2023.107170). Moreover, even if the intention to do a digital reconstruction of the Roman mosaic is said, the purpose would have been clearer if relevant research questions or hypothesis are mentioned in the manuscript so that the reader would be guided as to the study’s focus.

The technical details, for example software versions and settings, as well as equipment models, can really improve reproducibility. These data will help to gain a deeper and more practical understanding of the studied work process and its use in similar projects.

The discussion on the issues in data gathering and processing, which may arise e.g. when the mosaic is damaged in some places or missing, can enrich readers’ perception of the more practical aspects of mosaic restoring.

Although color management is mentioned, describing what is used for accurate colour matching and detailing the ways color is mapped to the original mosaic presentation can make the text stronger.

Considering the challenges and restrictions faced during this reconstruction period and unveiling any assumptions would make the results of the study more pragmatic and balanced.

Indicating research areas targeted for further research and possible enhancements to the methodology proposed for reconstruction will benefit the continuation and enrichment of the research in this field of study.

Author Response

Respectful Reviewer,

We are sincerely grateful for your constructive comments of your review that have enabled us to improve the written presentation of our research. Please find enclosed all the detailed explanations and description of the improvements that were added in our paper.  Each comment is in sequence explained (in red colour) and the corrections and improvements are red marked in the text of the paper. 

The paper was considering your comments corrected as follows:

Reviewer’s comment:

The article is relevant and interesting. Revision has to be done for the paper to publish it. I would really appreciate if the authors adjust their paper in the specified places. The beginning presents the historical context but could enhance the introductory part with a clearer connection of the significance of the mosaic's digital reconstruction and its addition to current research and technology in the field. In addition, some other reconstruction method are suggested (https:(https://doi.org/10.1016/j.enggeo.2023.107170). Moreover, even if the intention to do a digital reconstruction of the Roman mosaic is said, the purpose would have been clearer if relevant research questions or hypothesis are mentioned in the manuscript so that the reader would be guided as to the study’s focus.

Authors’ answer:

In the introduction 11 additional references were added giving more focused state of the art in the field included the paper (proposed by the Reviewer 1), and 1 additional reference in the Experimental part. Accordingly, Introduction was considerably restructured, marked in red in the manuscript.

The added references are: 7-8, 10, 14-17, 19, 25-28, 32

Giannotta, M.; Gabellone, F. New Data from Buried Archives and 3D Reconstruction The Late Roman Mosaic in Otranto (Italy).  In Proceedings of the 20th International Conference on Cultural Heritage and New Technologies - CHNT 20, Vienna 2015.

Lavric, M. 3D reconstruction of a balneum in a Roman Villa Rustica, Mošnje, Slovenia. In Conference: 2015 Digital Heritage 2015, pp. 727–730.

Ajioka, O.; Hori, Y. (2014). Application of SFM and laser scanning technology to the description of mosaics piece by piece. In ISPRS - International Archives of the Photogrammetry, Remote Sensing and Spatial Information Sciences 2014; XL-5, pp. 23–28.

Gabellone F.; Chiffi M.; Tanasi D.; Decker M. Integrated Technologies for Indirect Documentation, Conservation and Engagement of the Roman Mosaics of Piazza Armerina (Enna, Italy). In Proceedings of the 2nd International and Interdisciplinary Conference on Image and Imagination, Advances in Intelligent Systems and Computing 2020; Editor 1 Cicalò, E.; Springer; Nature Switzerland; vol 1140, pp. 10161028.

Fioretti, G.; Acquafredda P.; Silvia Calò, S.; Cinelli, M.; Germanò, G.; Laera, Moccia, A.  Study and Conservation of the St. Nicola's Basilica Mosaics (Bari, Italy) by Photogrammetric Survey: Mapping of Polychrome Marbles, Decorative Patterns and Past Restorations. Studies in Conservation 2020; 65(3), pp. 160171.

Fang, K.; Zhang, J.; Tang, H.; Hu, X.; Yuan, H.; Wang; X.; Pengju An, P.; Ding, B. A quick and low-cost smartphone photogrammetry method for obtaining 3D particle size and shape. Engineering Geology 2023, 322, 107170.

Pozzo D.; Scala B.; Adami A. Geometry and information for the preservation of a Roman Mosaic through HBIM Approach. In ISPRS Annals of the Photogrammetry, Remote Sensing and Spatial Information Sciences 2021; VIII-M-1-2021; pp. 7379.

Gherardini, F.; Santachiara, M.; Leali, F. Enhancing heritage fruition through 3D virtual models and augmented reality: an application to Roman artefacts. Virtual Archaeology Review 2019; 10(21), pp. 67–70.

Felicetti, F.; Albiero, A.; Gabrielli, R.; Pierdicca, R.; Paolanti, M.; Zingaretti, P.; Malinverni, E. S. Automatic Mosaic Digitalization: a Deep Learning approach to tessera segmentation. In Metrology for Archaeology and Cultural Heritage (MetroArchaeo), Cassino, Italy, 2018, pp. 132–136.

Ghosh, M.; Obadilluah, S.M; Gherardini, F.; Zdimalova, M. Classification of Geometric Forms in Mosaics Using Deep Neural Network. Journal of Imaging 2021; 7(8), pp. 149.

Gil, F. A.; Gomsi, J. M.; Pérez, M. Reconstruction Techniques for Image Analysis of Ancient Islamic Mosaics. The International Journal of Virtual Reality 2009; 8(3), pp. 512.

Lo Brutto, M.; Dardanelli, G. Vision metrology and Structure from Motion for archaeological heritage 3D reconstruction: a Case Study of various Roman mosaics. ACTA IMEKO 2017. 6(3), pp. 35.

InterArch-Steiermark. Available on line: URL http://www.interarch-steiermark.eu/sl/podatkovna_zbirka/predmeti/podrobnosti.html?item=1aebf0cc-7ad8-11e2-8b5a-e8393528f4bc (accessed on 25.3.2024).

In the Introduction aa research question was added (beside some additional explanatory about the goals of the research), that is discussed in the Conclusions:

  1. Introduction: lines 237-249

The Roman mosaic with the scene of the abduction of Europe is one of the most important mosaics found in the area of today's Slovenia, not only due to its quality, but also because of its still relevant message. The scene is connected with the myth of the colonization of our continent. That is also why we decided to make a graphic reconstruction of the entire mosaic. In addition to digital reconstruction, the steps involved in the digital reconstruction of the mosaic were systematically described, taking into account all phases of mosaic processing that are important for understanding the reconstruction of the mosaic.

Before starting the research, we posed the research question of "whether a complete reconstruction of the mosaic with the scene of the abduction of Europa, which is relatively well preserved and whose digital completion is predictable, can provide a visual experience, an interpretation in 3D and a starting point for further research and use with which the museum visitor can also interact".

  1. Conclusions: lines 535-542 and 551-555

Based on the aforementioned, we answer affirmatively to the research question stated at the beginning of the research. The research of existing sources (written and pictorial) about the mosaic and the digital reconstruction of the mosaic in 1:1 scale enables a comprehensive visual experience, an interpretation in interactive 3D animation with the aim of using it for exhibition purposes and as a starting point for further archaeological research.

Reviewer’s comment:

The technical details, for example software versions and settings, as well as equipment models, can really improve reproducibility. These data will help to gain a deeper and more practical understanding of the studied work process and its use in similar projects.

Authors’ answer:

Technical details of the phases of the workflow were added in the manuscript. Experimental part included the experimental framework (experimental diagram)

Figure 3. Reconstruction workflow.

3.1 Preliminary studies: lines 288-306, information about data gathering.

3.2. Photogrammetry: lines 317-336, software version, technical details of the procedure.

3.3. Color management: lines 372-387, software version, technical details of the procedure.

3.5. Interpretative 3D animation: lines 419-427, animation approatches.

Reviewer’s comment:

The discussion on the issues in data gathering and processing, which may arise e.g. when the mosaic is damaged in some places or missing, can enrich readers’ perception of the more practical aspects of mosaic restoring.

Authors’ answer:

Details and explanations about data gathering and processing of the mosaic in digital form were added

3.1. Preliminary studies: lines 288-306

  1. Results and Discussion, 4.1. Raster reconstruction: lines 430-496

Reviewer’s comment:

Although color management is mentioned, describing what is used for accurate colour matching and detailing the ways color is mapped to the original mosaic presentation can make the text stronger.

Authors’ answer:

Subsection 3.3. Color management was upgraded with the procedure of defining the representative colours of the mosaic (lines 358-369) and additional information about the colour management and the procedure were added in the experimental part

3.3. Colour management, lines: 372-387

Reviewer’s comment:

Considering the challenges and restrictions faced during this reconstruction period and unveiling any assumptions would make the results of the study more pragmatic and balanced.

Authors’ answer:

Discussion of the results was completely rewritten giving the explanations about the procedure, assumptions about the usefulness of the digital mosaic’s reconstruction. Additional explanations were added also in Conclusions giving the observations about the importance of the mosaic reconstruction

4.1. Raster reconstruction Lines 434-496

  1. Conclusions, lines 535-552 and 570-584

Reviewer’s comment:

Indicating research areas targeted for further research and possible enhancements to the methodology proposed for reconstruction will benefit the continuation and enrichment of the research in this field of study.

Authors’ answer:

The possibility of further research was presented in the Conclusions.

  1. Conclusions, lines 535-552 and 570-584

The presented study opens up possibilities for further research and deepening of the understanding of the meaning of the digital reconstruction of the mosaic. We are currently exploring the possibility of supplementing the methodology with AI, which we have used in the reconstruction of other mosaics, but which did not provide satisfactory results. The AI tools required an extremely large number of explanatory instructions, but these did not help to adequately fill in the missing parts of the mosaic. Moreover, improving the colour management process could be done even more specifically at the level of determining the colours of individual groups of tesserae belonging to a particular tile type. This would be based on the colour palettes used by geologists by taking into account the stone types. For this procedure, the geologist would have to determine the types of tesserae. Determining the origin would indicate the origin of the stones and thus also represent the process of mosaic production. We estimated that this process would be very time consuming and costly to implement, which in our opinion would not outweigh the quality and accuracy of the final digital reproduction for the purpose of presentation at the exhibition, however it would possibly aim the quality of further archaeological studies of the mosaic.

The Authors

Reviewer 2 Report

Comments and Suggestions for Authors

~line 50 - Because detailed description is already here, I would like to see a picture showing the mosaic.
line 115 - typo in et al.
line 195 - typo in Brg

Section 1
- Despite literature review about the mosaic reconstruction being convincing, the total number of references is rather low (23 in total). I recommend to extend their list by at least 10.
- Structure of section 1 is incorrect. Subsection 1.1 cannot be the only one subsection. Please consider making the subsection 1.1 to be section 2 that is outside the section 1. Section 2 will provide the literature review on the main topic of the article, which is good. Some information on the literature review protocol would be valuable as well. The remaining part of section 1 will be the introduction. It might be used to extend the list of references. What is even more important the scientific goal should be clearly highlighted - it might be done at the end of section 1.
- After splitting 1 into 2 separate sections, other sections should be re-enumerated.

Materials and methods:
- Please make it section 3
- Lines 231-252 the current style of punctuations is improper. They should be introduced somehow. Giving them just a caption is not enough. I believe that a reader deserves some technical and organizational/methodological details about the procedure. Currently they are missing or linking them with proper subsection is not straightforward. Moreover, by looking on the punctuations, the procedure looks enough complicated to be drawn as a diagram. In order to address all of these I would use another subsection at the beginning of section Materials and methods.

Lines 253-265 - these lines should be moved to the introduction

Line 291 - parenthesis in not closed

Line 293 - big empty space present

Lines 382-383 - make it one paragraph.

Figure 10 - I would give a number for each step (each part of Fig.10) to highlight the order of steps.

General
-For me stating the clear research/utilitarian goal, and conducting the text narracy accordingly to that goal, is missing.

Author Response

REVIEW 2      Applied Sciences (ISSN 2076-3417)

Manuscript ID: applsci-2902372

Title: Graphic Reconstruction of Roman Mosaic with the Scene with the Abduction of Europa

Authors: Gregor Oštir , Dejana Javoršek , Primož Stergar , Tanja Nuša Kočevar , Aleksandra Nestorović , Helena Gabrijelčič Tomc *

Respectful Reviewer,

We are sincerely grateful for your constructive comments of your review that have enabled us to improve the written presentation of our research. Please find enclosed all the detailed explanations and description of the improvements that were added in our paper.  Each comment is in sequence explained (in red colour) and the corrections and improvements are red marked in the text of the paper. 

The paper was considering your comments corrected as follows:

Reviewer’s comment:

~line 50 - Because detailed description is already here, I would like to see a picture showing the mosaic.

Authors’ answer:

Figures 1 and 2 with the associated text were moved to Introduction, where the meaning of the mosaic is described. In the corrected manuscript the lines are 54-65.

Reviewer’s comment:

line 115 - typo in et al.
line 195 - typo in Brg

Authors’ answer:

Corrected in the text of the manuscript. In the corrected manuscript the lines are 167 and 236

Line 176: " With the aim to reconstruct the Great Pitiunt Fortress Glazov et al. [23]"

Line 235: "Europa that was found in Ptuj, Zgornji Breg and is exhibited”

Reviewer’s comment:

Section 1
- Despite literature review about the mosaic reconstruction being convincing, the total number of references is rather low (23 in total). I recommend to extend their list by at least 10.

Authors’ answer:

In the introduction 11 additional references were added giving more focused state of the art in the field included the paper (proposed by the Reviewer 2), and 1 additional reference in the Experimental part. Accordingly, Introduction was considerably restructured, marked in red in the manuscript. In total now there are 35 references, References lines 600-677.

The added references are: 7-8, 10, 14-17, 19, 25-28, 32

Giannotta, M.; Gabellone, F. New Data from Buried Archives and 3D Reconstruction The Late Roman Mosaic in Otranto (Italy).  In Proceedings of the 20th International Conference on Cultural Heritage and New Technologies - CHNT 20, Vienna 2015.

Lavric, M. 3D reconstruction of a balneum in a Roman Villa Rustica, Mošnje, Slovenia. In Conference: 2015 Digital Heritage 2015, pp. 727–730.

Ajioka, O.; Hori, Y. (2014). Application of SFM and laser scanning technology to the description of mosaics piece by piece. In ISPRS - International Archives of the Photogrammetry, Remote Sensing and Spatial Information Sciences 2014; XL-5, pp. 23–28.

Gabellone F.; Chiffi M.; Tanasi D.; Decker M. Integrated Technologies for Indirect Documentation, Conservation and Engagement of the Roman Mosaics of Piazza Armerina (Enna, Italy). In Proceedings of the 2nd International and Interdisciplinary Conference on Image and Imagination, Advances in Intelligent Systems and Computing 2020; Editor 1 Cicalò, E.; Springer; Nature Switzerland; vol 1140, pp. 10161028.

Fioretti, G.; Acquafredda P.; Silvia Calò, S.; Cinelli, M.; Germanò, G.; Laera, Moccia, A.  Study and Conservation of the St. Nicola's Basilica Mosaics (Bari, Italy) by Photogrammetric Survey: Mapping of Polychrome Marbles, Decorative Patterns and Past Restorations. Studies in Conservation 2020; 65(3), pp. 160171.

Fang, K.; Zhang, J.; Tang, H.; Hu, X.; Yuan, H.; Wang; X.; Pengju An, P.; Ding, B. A quick and low-cost smartphone photogrammetry method for obtaining 3D particle size and shape. Engineering Geology 2023, 322, 107170.

Pozzo D.; Scala B.; Adami A. Geometry and information for the preservation of a Roman Mosaic through HBIM Approach. In ISPRS Annals of the Photogrammetry, Remote Sensing and Spatial Information Sciences 2021; VIII-M-1-2021; pp. 7379.

Gherardini, F.; Santachiara, M.; Leali, F. Enhancing heritage fruition through 3D virtual models and augmented reality: an application to Roman artefacts. Virtual Archaeology Review 2019; 10(21), pp. 67–70.

Felicetti, F.; Albiero, A.; Gabrielli, R.; Pierdicca, R.; Paolanti, M.; Zingaretti, P.; Malinverni, E. S. Automatic Mosaic Digitalization: a Deep Learning approach to tessera segmentation. In Metrology for Archaeology and Cultural Heritage (MetroArchaeo), Cassino, Italy, 2018, pp. 132–136.

Ghosh, M.;  Obadilluah, S.M; Gherardini, F.; Zdimalova, M. Classification of Geometric Forms in Mosaics Using Deep Neural Network. Journal of Imaging 2021; 7(8), pp. 149.

Gil, F. A.; Gomsi, J. M.; Pérez, M. Reconstruction Techniques for Image Analysis of Ancient Islamic Mosaics. The International Journal of Virtual Reality 2009; 8(3), pp. 512.

Lo Brutto, M.; Dardanelli, G. Vision metrology and Structure from Motion for archaeological heritage 3D reconstruction: a Case Study of various Roman mosaics. ACTA IMEKO 2017. 6(3), pp. 35.

InterArch-Steiermark. Available on line: URL http://www.interarch-steiermark.eu/sl/podatkovna_zbirka/predmeti/podrobnosti.html?item=1aebf0cc-7ad8-11e2-8b5a-e8393528f4bc (accessed on 25.3.2024).

Reviewer’s comment:

- Structure of section 1 is incorrect. Subsection 1.1 cannot be the only one subsection. Please consider making the subsection 1.1 to be section 2 that is outside the section 1. Section 2 will provide the literature review on the main topic of the article, which is good. Some information on the literature review protocol would be valuable as well. The remaining part of section 1 will be the introduction. It might be used to extend the list of references. What is even more important the scientific goal should be clearly highlighted - it might be done at the end of section 1.

- After splitting 1 into 2 separate sections, other sections should be re-enumerated.

Authors’ answer:

Section 1 was considerably restructured (without subsections) and divided to two sections: 1. Introduction, lines 29-86, presenting the studied mosaic, and 2. Examples of Mosaic reconstruction, which was considerably restructures and added with 12 references about mosaic reconstruction, lines 87-231.

The scientific goals with the research question (lines 237-249) were added in the paragraphs defining the aims of the research 230-266

“The goals of the presented research were metrically accurate reconstruction, colour-accurate reproduction and the interpretation of the mosaic in interpretative 3D animated presentation, of the remains of an ancient Roman mosaic with scene of the abduction of Europa that was found in Ptuj, Zgornji Breg and is exhibited in the Universalmuseum Joanneum in Graz (Austria). The Roman mosaic with the scene of the abduction of Europe is one of the most important mosaics found in the area of today's Slovenia, not only due to its quality, but also because of its still relevant message. The scene is connected with the myth of the colonization of our continent. That is also why we decided to make a graphic reconstruction of the entire mosaic. In addition to digital reconstruction, the steps involved in the digital reconstruction of the mosaic were systematically described, taking into account all phases of mosaic processing that are important for understanding the reconstruction of the mosaic.

Before starting the research, we posed the research question of "whether a complete reconstruction of the mosaic with the scene of the abduction of Europa, which is relatively well preserved and whose digital completion is predictable, can provide a visual experience, an interpretation in 3D and a starting point for further research and use with which the museum visitor can also interact".”

The sections were renumbered after splitting (1. Introduction and 2. Examples of Mosaic reconstruction) without subsections.

Reviewer’s comment:

Materials and methods:
- Please make it section 3
- Lines 231-252 the current style of punctuations is improper. They should be introduced somehow. Giving them just a caption is not enough. I believe that a reader deserves some technical and organizational/methodological details about the procedure. Currently they are missing or linking them with proper subsection is not straightforward. Moreover, by looking on the punctuations, the procedure looks enough complicated to be drawn as a diagram. In order to address all of these I would use another subsection at the beginning of section Materials and methods.

Authors’ answer:

Materials and methods are numbered as Section 3, line 276.

At the beginning of the section Materials and methods the framework as diagram was added.

The experimental part of the graphic reconstruction of the mosaic was carried out by a multidisciplinary team. The team consisted of experts in archaeology, photogrammetric acquisition, graphic design and visualization, and colour management, who determined the workflow of the reconstruction. The reconstruction workflow was divided in five phases: 1. preliminary studies; 2. photogrammetry; 3. raster reconstruction; 4. colour management and printing and 5. 3D interpretative animation as is with the subphases presented in Figure 3.

Please see corrected manuscript

Figure 3. Reconstruction workflow.

All the sequent subsections were considerably upgraded, so that in our oppinion is easier to follow the framework. Moreover, we added also additiona explanations in the subsections

3.1. Preliminary studies, lines 287-315

3.2. Photogrammetry, lines 316-256

3.3. Colour management, lines 357-390

3.5. Interpretative 3D animation, lines 418-427

Reviewer’s comment:
Lines 253-265 - these lines should be moved to the introduction
Line 291 - parenthesis in not closed

Line 293 - big empty space present

Lines 382-383 - make it one paragraph.

Authors’ answer:

Lines 253-265 were oved to Introduction, now lines 54-65

Line 291 - parenthesis is closed, now line 364, (4x i1Pro 2, X-Rite)

Line 293 – there is no more big empty space, now lines 265-266….”are colour 1 white (CIELab = 61.245, -0.945, 5.014), colour 2 grey (CIELab = 52.612, 1.432, -3.792), colour 3 black and dark grey”

Reviewer’s comment:

Figure 10 - I would give a number for each step (each part of Fig.10) to highlight the order of steps.

Authors’ answer:

In Figure 10 – now Figure 11 nubmers were added and additional explanatory text, lines 434-452

The mosaic was relatively well preserved, with preserved essential elements of the mosaic, such as borders, figures, repeating pattern fields, which facilitated the transfer of pictorial information and preserved surfaces to non-preserved ones, with the aim of completing it as authentically as possible.

Please see corrected manuscript

Figure 11. Key steps of raster reconstruction.

Since the mosaic scheme is based on the perfect symmetry of the motifs (Figure 11 a.), initially the right side of the mosaic was reconstructed by mapping the upper (original) half (Figure 11 b.). One of the four triangles, whose vertices touch the inner circle, was supplemented and the frame border with multi-colored double strand guilloche, the black band and a white frame band were added (Figure 11 c., 11 d.). When this part was done, more than half of the mosaic was reconstructed (Figure 11 e.). Then the preserved part of the mosaic on the left side was mapped (Figure 11 f.) and the missing triangle was filled the same way as the one on the right side (Figure 11 g., 11 h., 11 i.). At the same time, the missing part of the multi-colored double strand guilloche of the central field with a figural scene was added (Figure 11 g.). The final product was improved, and errors of the mosaic were corrected that would otherwise have been repeated. We also cleaned up the damage on the white frame band (Figure 11 j., 11 k., 11 l.).

Reviewer’s comment:

-For me stating the clear research/utilitarian goal, and conducting the text narracy accordingly to that goal, is missing.

Authors’ answer:

Basically, the main manuscript has been significantly upgraded so that it is easier to follow the research protocol and work phases, i.e. the research aim has been more clearly defined, including the research question. The references have been significantly improved (12 additional references in section 2); an explanatory diagram has been added to show the framework; the experimental part has been restructured by adding the technical settings, protocols, etc. The Results and Discussion section has been rewritten to give the reader a more detailed insight into the scientific significance of the research results. In addition, some comments have been added in the conclusions to answer the research question and for further research.

The Authors

Reviewer 3 Report

Comments and Suggestions for Authors

The digital reconstruction of a Roman mosaic that was found in Ptuj, Slovenia, during the second and fourth halves of the third and fourth centuries, and depicts the European Renaissance is examined in this study. Using an interdisciplinary approach that incorporates archeology, photogrammetry, graphic design, and visualization, the team created a 2D reconstruction based on photogrammetric data and color management. For the mosaic's presentation, an interpretative 3D animation was subsequently made using this reconstruction. The initiative offers new prospects for the study and preservation of cultural assets without needing invasive adjustments to the original, highlighting the importance of multidisciplinary cooperation and the application of digital technology in the preservation and exhibition of cultural property.

The work, although well presented, does not bring a real innovation in the field of image reconstruction, in this case of mosaic images. Within the article, there is an application of a technique, the innovation brought by this experimentation is not explained, nor is the entire image reconstruction step explained in detail. In general, the entire article should be revised, from the poorly explained objectives in the introductory part, and the study of related work should also be improved, to better frame where one wants to place oneself with one's research. Finally, there is a lack of commentary on the results obtained, and a chapter on the validation of both the result and the process. I also find the chapter "3.1. Interpretative video animation", because neither is it commented on nor explained why a video reconstruction is used.

We recommend exploring artificial intelligence techniques to support the work and a comparison of the results obtained as an example. This is why I advise the authors to continue exploring in this area, and resubmit the work at a later date when it is more mature.

Comments on the Quality of English Language

From a grammatical and reading point of view, there are no particular problems.

Author Response

REVIEW 3      Applied Sciences (ISSN 2076-3417)

Manuscript ID: applsci-2902372

Title: Graphic Reconstruction of Roman Mosaic with the Scene with the Abduction of Europa

Authors: Gregor Oštir , Dejana Javoršek , Primož Stergar , Tanja Nuša Kočevar , Aleksandra Nestorović , Helena Gabrijelčič Tomc *

Respectful Reviewer,

We are sincerely grateful for your constructive comments of your review that have enabled us to improve the written presentation of our research. Please find enclosed all the detailed explanations and description of the improvements that were added in our paper.  Each comment is in sequence explained (in red colour) and the corrections and improvements are red marked in the text of the paper. 

The paper was considering your comments corrected as follows:

Reviewer’s comment:

The work, although well presented, does not bring a real innovation in the field of image reconstruction, in this case of mosaic images. Within the article, there is an application of a technique, the innovation brought by this experimentation is not explained, nor is the entire image reconstruction step explained in detail. In general, the entire article should be revised, from the poorly explained objectives in the introductory part, and the study of related work should also be improved, to better frame where one wants to place oneself with one's research. Finally, there is a lack of commentary on the results obtained, and a chapter on the validation of both the result and the process. I also find the chapter "3.1. Interpretative video animation", because neither is it commented on nor explained why a video reconstruction is used.

Authors’ answer:

Basically, the main manuscript has been significantly upgraded so that it is easier to follow the research protocol and work phases, i.e. the research aim has been more clearly defined, including the research question. The references have been significantly improved (12 additional references in section 2); an explanatory diagram has been added to show the framework; the experimental part has been restructured by adding the technical settings, protocols, etc. The Results and discussion section has been rewritten to give the reader a more detailed insight into the scientific significance of the research results. In addition, some comments have been added in the conclusions to answer the research question and for further research. We have added about 4 pages of additional clarifications and explanations, technical descriptions and justification of the meaning of the mosaic in Slovene to the contribution.

 We believe that on this basis, the work makes a researched contribution on the example of the Roman mosaic, which is important in the field of Slovenia and Austria. It is a unique digital reconstruction of its kind in the field, which is also interpreted into exhibition halls with the aim of bringing visitors closer. Please see the corrected manuscript for the corrections and upgrades.

Examples of the upgraded text:

  1. Introduction, Goals, lines 237-249

The Roman mosaic with the scene of the abduction of Europe is one of the most important mosaics found in the area of today's Slovenia, not only due to its quality, but also because of its still relevant message. The scene is connected with the myth of the colonization of our continent. That is also why we decided to make a graphic reconstruction of the entire mosaic. In addition to digital reconstruction, the steps involved in the digital reconstruction of the mosaic were systematically described, taking into account all phases of mosaic processing that are important for understanding the reconstruction of the mosaic.

Before starting the research, we posed the research question of "whether a complete reconstruction of the mosaic with the scene of the abduction of Europa, which is relatively well preserved and whose digital completion is predictable, can provide a visual experience, an interpretation in 3D and a starting point for further research and use with which the museum visitor can also interact".

  1. Materials and methods

The experimental part of the graphic reconstruction of the mosaic was carried out by a multidisciplinary team. The team consisted of experts in archaeology, photogrammetric acquisition, graphic design and visualization, and colour management, who determined the workflow of the reconstruction. The reconstruction workflow was divided in five phases: 1. preliminary studies; 2. photogrammetry; 3. raster reconstruction; 4. colour management and printing and 5. 3D interpretative animation as is with the subphases presented in Figure 3.

Please see the corrected manuscript

Figure 3. Reconstruction workflow.

3.1. Preliminary studies

    The analysis of the surviving pictorial and written sources on the mosaic was carried out by an expert in the archaeology of the Roman period in question. The sources were evaluated qualitatively with regard to their archaeological relevance, using information about the time of the mosaic's creation (social, residential, economical organisation) and the starting points of the mosaic's creation. Sources were preserved part of the original mosaic, database InterArch-Steiermark [32], data base of the Universalmuseum Joanneum (hand drawing of the mosaic on foil in the ratio 1:1), publications of the mosaic with photos, and drawings and floor plan of the architecture [1, 2, p. 4, 5; 3 p. 599, 601, 602].

4.1. Raster reconstruction

 Key steps of raster reconstruction are presented in Figure 11. The result is a digital graphic reconstruction of the complete mosaic of the abduction of Europa, including the missing parts, as shown in Figure 12. Most of the mosaic is based on geometric patterns that are typical of the period in which the mosaic was made and are repeated evenly throughout the mosaic. The mosaic was relatively well preserved, with preserved essential elements of the mosaic, such as borders, figures, repeating pattern fields, which facilitated the transfer of pictorial information and preserved surfaces to non-preserved ones, with the aim of completing it as authentically as possible.

Please see the corrected manuscript

Figure 11. Key steps of raster reconstruction.

Since the mosaic scheme is based on the perfect symmetry of the motifs (Figure 11 a.), initially the right side of the mosaic was reconstructed by mapping the upper (original) half (Figure 11 b.). One of the four triangles, whose vertices touch the inner circle, was supplemented and the frame border with multi-colored double strand guilloche, the black band and a white frame band were added (Figure 11 c., 11 d.). When this part was done, more than half of the mosaic was reconstructed (Figure 11 e.). Then the preserved part of the mosaic on the left side was mapped (Figure 11 f.) and the missing triangle was filled the same way as the one on the right side (Figure 11 g., 11 h., 11 i.). At the same time, the missing part of the multi-colored double strand guilloche of the central field with a figural scene was added (Figure 11 g.). The final product was improved, and errors of the mosaic were corrected that would otherwise have been repeated. We also cleaned up the damage on the white frame band (Figure 11 j., 11 k., 11 l.).

Please see the corrected manuscript

Figure 12. The final digital graphic reconstruction of the entire mosaic.

 The work was progressive, each step of the digital addition made it possible to think about the next step, both from the archeological point of view to understand the meaning of each component of the mosaic, and from the point of view of graphic interventions and editing of the image information with the aim of achieving the visual communication of the mosaic. During the work process, we had the greatest challenges in identifying the darker band in the upper part after we had assembled the reconstruction of the photogrammetric images. We realized that due to the lighting (light through the windows) in the museum where the mosaic is exhibited, this upper part is regularly exposed to slight light influences and is therefore also slightly darker. In addition, in the final reconstruction, when visualizing the entire mosaic, digital image processing interventions are known to occur in areas with patches of visual artifacts that are unevenly distributed across the surface of the mosaic and are more noticeable in areas where there are fewer colored elements of the mosaic. Figure 13 shows on the left the shadow "cast" on the mosaic by the ceiling of the museum, on the surface of which we discovered a darker band of tesserae even after additional illumination of the mosaic, and on the right visual artifacts on the surface of the lower left part of the mosaic, which was reconstructed entirely from the image information of the preserved parts.

Please see the corrected manuscript for all the corections marked in red.

Reviewer’s comment:

We recommend exploring artificial intelligence techniques to support the work and a comparison of the results obtained as an example. This is why I advise the authors to continue exploring in this area, and resubmit the work at a later date when it is more mature.

Authors’ answer:

We are very grateful for this comment. Of course, we are also investigating the use of AI, especially on mosaics that are much worse preserved than those represented in this study. For the mosaic in question, with scenes from Europe, the process did not prove useful, mainly because of the need to show the reconstruction process for exhibition purposes. We have included this in the paper as a possibility for further research.

  1. Results and discussion

Lines 503-506

In our opinion such a systematic presentation of the working layers of the reconstruction would not be possible if fully automated and algorithmically controlled processes were used in the reconstruction. In this way, the digital composition of the mosaic was brought closer to the exhibition visitors in a visual and tangible way.

  1. Conclusions

Lines 570-575

The presented study opens up possibilities for further research and deepening of the understanding of the meaning of the digital reconstruction of the mosaic. We are currently exploring the possibility of supplementing the methodology with AI, which we have used in the reconstruction of other mosaics, but which did not provide satisfactory results. The AI tools required an extremely large number of explanatory instructions, but these did not help to adequately fill in the missing parts of the mosaic.

The Authors

Round 2

Reviewer 3 Report

Comments and Suggestions for Authors

In reviewing the text, I have identified a number of areas that, in my opinion, need special attention in order to improve the clarity and validity of scientific communication. Accordingly, I will list my main suggestions:

the section immediately following the introduction is devoted to the discussion of photogrammetry in a general context. I propose to eliminate or significantly reduce this part, focusing instead on techniques applied specifically to mosaics. It is essential to clearly outline why these technologies were selected for study and how they form the basis of the proposed approach.

 After the description of the relevant techniques, it is crucial to explicitly motivate the innovative content of the study in relation to these techniques. In this way, the added value of the presented research, which I see as weak at the moment, can be more precisely outlined.

The paragraph beginning at line 233 extends to line 275 without interruption. I would consider starting a new paragraph at line 233 to improve readability. Or, the information contained in lines 233-244 could be appropriately moved to the introductory section, thus improving the overall structure of the text.

There is also a discrepancy in the order in which techniques are presented; in particular, vectorisation seems to be discussed before orthophotos. This order appears inverted from the intended logical flow. I would suggest reconsidering this sequence to more accurately reflect the methodological process followed.

Take into account in the state-of-the-art study also artificial intelligence applications to support such techniques, and also in the part of possible future developments, such as the creation of a dedicated mosaic dataset.

 Finally, the document does not specify the Ground Sample Distance (GSD) of the orthophoto, a technical detail relevant to understanding the quality and accuracy of the images analysed. The inclusion of such technical details is crucial for a complete evaluation of the work.

Overall, the work has an interesting application, but the clarity and effectiveness of the communication could benefit greatly from the suggested changes. The innovative part of the work is missing in comparison to applications already present in the scientific field regarding image reconstruction.

Comments on the Quality of English Language

The work has long periods, for language there are no major problems.

Author Response

Dear Reviewer,

We are sincerely grateful for your constructive comments of your review that have enabled us to improve the written presentation of our research. Please find enclosed all the detailed explanations and description of the improvements that were added in our paper.  Each comment is in sequence explained (in red- 1st corrections and blue colour-2nd corrections) and the corrections and improvements are red marked in the text of the paper. 

The corrected version of the manuscript contains optimised references (2 were added, 1 was removed), which are now explained as they show the significance for the mosaic reconstruction and were restructures taking into account the meaning for the theory context. On research design: added the explanation of why the vector line drawing was created before photogrammetry (see explanation below). The suggested explanations have been added in the methodology (photogrammetry) and the conclusions have been upgraded with the possibilities of further implementation of AI in the framework. The English language has been corrected so that terms are used in line with grammatical appropriateness.

Round 3

Reviewer 3 Report

Comments and Suggestions for Authors

The authors responded to all suggested reviews satisfactorily. I congratulate the authors for the thorough revision work they have done.